# The light at the end of the tunnel? A systematic review of higher education student experiences of hope

Clio Berry[1]*, Nishi Acharya[1], Lucie Crowter[2]

1 Brighton and Sussex Medical School, University of Sussex, United Kingdom, 2 School of Psychology, University of Sussex, United Kingdom

* c.berry@bsms.ac.uk

**Data Availability Statement:** This study is a systematic review and meta-synthesis of published data. The data were those presented in the primary research reports reviewed. All the primary research

## Abstract

The most dominant model of hope is cognitive, in which hope is defined as goal-directed thinking, comprising self-agency and goal route identification. Nonetheless, competing theories about the fundamental nature of hope remain and further exploration of the construct is warranted. Little is known about whether the cognitive model aligns with how higher education students themselves think about hope. Understanding how "lay" populations conceptualise and experience psychological phenomena is as important as applying scientific theory. Personal beliefs impact on how people make sense of their life experiences and on their wellbeing. Research is specifically needed to explore the conceptualisation and experience of hope within diverse student populations. A systematic review was conducted to identify published scientific research and grey literature presenting qualitative accounts of hope from higher education students. A qualitative meta-synthesis of the eight eligible reports was conducted using thematic analysis and synthesis approaches to identify conceptualisations of hope and its associations with mental health and well-being. Nine themes were identified, reflecting that hope is: fundamental; self-construal over time; goal-directed; cognitive-emotional-behavioural; connection; resilience; dynamic and reciprocal; the inverse of depression; positive. These findings were identified as reflecting conclusions in which at least a moderate level of confidence may be placed. These findings clearly align with the cognitive model of hope, but emphasise the additional facets of fundamentality, self-construal, and negative origins. The implications for higher education institutions include to promote growth mindsets, to support students to learn skills for identifying and pursuing goals, and to provide hope-enhancing interventions as part of their student support provision.

## Introduction

Hope has long been a topic of interest across disciplines including philosophy, theology, and psychology. Different psychological models have been proposed, with cognitive hope theory as the most dominant. Charles Snyder, an American psychologist, defined hope as goal-directed cognition, involving self-agency and the identification of specific pathways (routes) towards

reports are referenced within the paper. The data are available in these referenced reports.

**Funding:** The author(s) received no specific funding for this work.

**Competing interests:** The authors have declared that no competing interests exist.

achieving goals [1]. Goals are the focus of hope; they must be meaningful and realistic, but not so easily achievable that they would not occupy hopeful thinking [1]. Self-agency, sometimes called "willpower" or "the will", is the motivation and determination to reach towards one's goals and the belief that one can achieve them [1]. Pathways thinking, sometimes called "way-power" or "the ways", is the capacity to use planning and to identify one or more specific means to move towards reaching one's goals [1].

Snyder's hope theory is not without criticism. Some research has indicated that agency and pathways are not equally associated with hope, but instead that agency has more explanatory power [2]. Nonetheless, agentic individuals are not necessarily able to create pathways towards their desired future [2]. Other researchers have suggested that hope should be viewed as socio-emotional, as opposed to (or in addition to) a cognitive construct [3–5]. Aspinwall and Leaf [4] criticised Snyder's model for overlooking these aspects, especially neglecting the social context of an individual, which they argued significantly impacts both agency and pathways. Snyder's model offers some defence in that it suggests that hope arises in the context of a secure attachment relationship, that hope buffers against social adversity, and that emotion is closely interlinked with hopeful thinking and goal pursuit [1]. Nonetheless, competing theories about the fundamental nature of hope remain and further exploration of the construct is warranted. Research additionally has highlighted a need for increased exploration of hope within diverse populations, as people from different cultures and countries experience hope differently [6, 7].

Of particular interest is the experience of hope in the transition from adolescence to adulthood, and specifically considering the higher education context. The transition to adulthood is a critical period for developing a sense of identity and building a future [8, 9]. The transition to, and through, university reflects a period of marked mental health vulnerability, involving adjusting to a new social system while managing dynamic academic, financial, and romantic relationship stressors [10]. Increasing numbers of university students are reporting mental health problems and dropping out for mental health-related reasons [11–14], thus university student mental health is now a global public health concern. Hope is a relevant lens through which to examine university student experiences. In the higher education context, much research now suggests that hope is robustly associated with psychological, social, and academic functioning [15]. For example, hope is correlated with academic achievement, athletic performance, increased life satisfaction, increased wellbeing, better coping, and increased use of adaptive health behaviours [16–18]. Hope is also associated with lower likelihood of academic misconduct and reduced suicidality [17, 19].

One important vantage point through which to explore the nature of hope is phenomenology, seeking to add to the limited understanding of how individuals experience and make sense of hope [7]. How people make sense of psychological phenomena, embedded within ordinary language, is as important to understanding as scientific theory [20]. People's "lay" beliefs play a significant role in their daily lives through shaping how they make sense of their past, present, and future experiences, how they evaluate their growth and progress through life, and the appraisals they make about their capability and potential [21]. Lay theories also affect how people understand and react to the actions of others [22]. Therefore, there are multiple ways in which lay theories of hope can influence wellbeing [23, 24]. Moreover, if lay theories differ from the scientific conceptualisations of hope, then these theoretical models will fail to generalise accurately with respect to predicting student outcomes.

Studies on hope involving young people with mental health and/or social problems in the UK and internationally have suggested that the cognitive model of hope as goal-directed self-agency and pathways thinking aligns with how they conceptualise and experience hope [25–27]. Yet young people do additionally emphasise the relational aspects of hope, the impact of trauma and adversity, and hope's connection to spirituality [25, 27, 28]. More generally, there

is a lack of exploratory and inductive higher education research designed to establish more nuanced understandings of hope [15]. Thus, collating evidence on how higher education students understand and experience hope, and its relationship with mental health and wellbeing, is a worthwhile endeavour. Such work can inform the development of optimal learning environments that maximise student wellbeing and positive functioning.

## Aim

We aimed to systematically identify and synthesise qualitative data on the higher education student experience of hope and its connections with mental health and wellbeing. The specific research questions were: How do higher education students conceptualise and experience hopefulness? How do higher education students perceived hopefulness to be associated with their mental health and wellbeing?

## Method

This study is a systematic review and meta-synthesis of qualitative data. The identification of data comprised a scientific evidence and grey literature search. The meta-synthesis used thematic synthesis and thematic analysis techniques as described by Thomas and Harden [29] and Braun and Clarke [30] respectively.

## Protocol and registration

The protocol was pre-registered in the International Prospective Register of Systematic Reviews (PROSPERO) register for systematic reviews, under registration number CRD42022369321 on 25/10/22.

## Data sources

Studies were first identified through a search of five electronic databases: Pubmed (MED-LINE), Web of Science, PsycINFO, CINAHL Plus and Scopus. The search terms (see Table 1) were determined using the SPIDER [31] framework and focused on hope, higher education students, and qualitative and mixed methodologies. Grey literature was then identified using the Open Grey database. Due to the lack of search filters, a smaller selection of key words was used, i.e., "hope" and "students". All searches were carried out during October and November 2022.

## Study selection

Studies retrieved by the scientific and grey literature searches were stored in a reference management software, Endnote, and uploaded onto Rayyan [32]. Retrieved studies were screened in a two-stage process; title-abstract and then full-text. Studies were included if they a) presented data on the hope experiences of higher education students aged 16 years or over in any stage and mode of study, b) used qualitative or mixed-method designs, and c) were published in the English language. Studies were excluded if they a) were only quantitative or b) contained no primary data (including systematic reviews, meta-analyses, letters to the editor, opinion and position papers).

At title-abstract stage, records were screened by NA, with a random 20% independently screened by CB. Studies were screened by both reviewers blindly on Rayyan and then the decisions were unmasked. The rate of conflict was acceptable at 14% of those double screened. In the full-text screening stage, records were screened by NA, again with 20% independently dual-screened by CB. A conflict of three studies (6.7%) was identified, which was acceptably

**Table 1. Scientific database search terms.**

| | |
|---|---|
| Sample | university* OR "higher education" OR "college*" OR "postgraduate*" OR "doctoral student*" OR "research student*" OR student* OR graduate* OR undergraduate* OR "young people" OR youth OR "young adult*" OR adolescen* OR "emerging adult*" OR trainee* OR candidate* OR layperson OR "lay people" OR laypeople |
| Phenomenon of Interest | hope/hopefulness |
| Design | interview* OR "focus group*" OR "qualitative survey*" OR "qualitative question*" OR "qualitative item*" OR "qualitative response*" |
| Evaluation | "hope experience*" OR "experience of hope*" OR "experience hope*" OR "experiencing hope*" OR "hopefulness experience*" OR "hopefulness analysis" OR "hopefulness definition*" OR<br>"understanding hope*" OR "understanding of hope*" OR "understandings of hope*" OR "hope analysis" OR "analysis of hope*" OR "hope meaning*" OR "hopefulness meaning" OR "meanings of hope*" OR "meaning of hope*" OR "concept of hope*" OR "concepts of hope*" OR "definition of hope*" OR "defining hope*" OR "definitions of hope*" OR "hope definition*" OR "hope concept*" OR "conceptualising hope*" OR "model of hope*" OR "beliefs about hope*" OR "belief about hope*" OR "sources of hope*" OR "source of hope*" OR<br>"hopefulness belief*" OR "phenomenology of hope*" OR "hope phenomenology" OR "hopefulness phenomenology" |
| Research type | qualitative OR "mixed method*" OR "mixed-method*" OR thematic OR interpretative OR IPA OR phenomenological OR "content analysis" OR "grounded theory" OR "narrative analysis" |

Note: Categories were combined as follows: [S] AND [PI OR E] AND [D OR R].

low. All conflicts were discussed until a consensus agreement was reached and a final list of records were obtained for inclusion in the systematic review.

## Data extraction

Key characteristics of eligible studies were extracted into a Microsoft Excel spreadsheet. Extracted data included first general study details; study author(s) and year of publication, country, study aim(s). Next, we extracted data on methodology (including methodological details, study type and design, study setting, epistemology, data collection tools, and type of data analysis). Finally, we extracted data on participants: participant selection (inclusion and exclusion criteria) and participant demographics (total number, age range, mean age, higher education course and institution, race/ethnicity, gender). These data are presented in Table 2. Finally, the qualitative data were extracted and analysed in a separate Microsoft Word document.

## Evidence synthesis

Elements of thematic synthesis [29] and analysis [30] were used to synthesise the data. Initially, stages one and two of the thematic analysis method were followed. All included papers were read thoroughly. Extracted qualitative data were then read multiple times to facilitate familiarisation. Next, inductive coding was carried out. Further coding was then performed to interpret the views of the study participants in-depth and find connections between codes. Next, stage two of the thematic synthesis and three of the thematic analysis method (searching for themes) were completed. Connections between codes were examined in detail to group codes and create themes. Central Organizing Concepts [30] were identified to create distinct themes which best captured the data patterns. The last steps involved stages four and five of thematic analysis (reviewing and defining themes). Themes were reviewed and discussed by the co-authors to refine the theme names and codes within these themes.

**Table 2. Study characteristics.**

| First author (date) | Country | Aim(s) | Methodology | Epistemology | Setting | Data collection tool(s) | Data analysis | Inclusion and exclusion (Incl, Excl) criteria | Sample size (Age range; mean (SD)) | Sample characteristics (Sex (%); Ethnicity N(%)) | Summary of results |
|---|---|---|---|---|---|---|---|---|---|---|---|
| Chamodraka (2008) | Canada | To explore hope from the perspective of clients who experienced increased hope during psychotherapy | Qualitative, grounded theory | Constructivist | Urban university counselling centre serving undergraduate and postgraduate students | Individual interviews | Grounded theory | Incl: Significant increase in hope over course of therapy, equating to increase of 6 points on State Hope Scale and average item score at least 5 out of 8, indicating mainly agreement | 18 (18–43; 23.7(5.86) | Female 72%, Male 28%; White 56%, 12% European-Canadian, 12% German, 6% Canadian, 6% South-Asian, 6% Chinese-Taiwanese, 6% Hungarian-Polish, 6% Mahgrebian | Three major sources of hope were identified: new outlook, improvement/ sense of accomplishment, and having a supportive space to talk. |
| De Pretto (2020) | Italy | To extend existing psychological knowledge on hope and to explore beyond its cognitive conceptualisation | Qualitative, phenomenological | Constructivist | University undergraduate psychology classroom | Qualitative survey | Thematic analysis | Incl: Italian psychology undergraduate students | 46 (96% 20–27, 4% 29–52; Unknown) | Female 78%, Male 22%; unknown | Students wrote about their most important hopes in life and identified sources of support |
| Hulme (1997) | USA | To investigate anticipatory consciousness and to explore the relationship between hope and despair and the motivation and desire to learn | Qualitative, grounded theory | Constructivist | University conference; campus sites | Group interviews; individual interviews and diary entries | Narrative analysis | Incl: University students randomly selected from two private universities, one faith-based | 32 (Unknown; Unknown) | Female 72%, Male 28%; Unknown | Two dimensions of hope were identified; concrete and transcendent |
| Jones (2015) | UK | To gain an understanding of what hope is, how it is fostered, and to explore how hope is related to resilience of youth in supported residences | Qualitative, phenomenology | Constructivist | Supported youth residences | Individual interviews | Interpretative Phenomenological Analysis | Incl: 18–25 year-olds who had lived in a childcare facility in The Bahamas for at least three consecutive years | 4* (22–23 years; Unknown) | Female 50%, Male 50%;75% Bahamian, 25% Haiti-Bahamian | Hope is faith in a better future and is an expectation. Hope is greatly affected by family support. |
| Lam (2017) | China | To investigate young adults' academic success and to explore the place of hope in the journey from academic failure to success | Mixed methods**, phenomenology | Constructivist | University campus | Individual interviews** | Interpretative Phenomenological Analysis** | Incl: Graduates of an alternative continuing education pathway, with high hope (score of 56 or more on Trait Hope Scale) and low functioning families (score of 2 or less on GFS scale of The Family Assessment Device) | 8** (19–22; 19.9(1.13)) | Female 50%, 50% Male; Unknown | Four themes regarding hope identified: adverse conditions, motivation of interest, positive thoughts, and relationship support |

*(Continued)*

**Table 2.** (Continued)

| First author (date) | Country | Aim(s) | Methodology | Epistemology | Setting | Data collection tool(s) | Data analysis | Inclusion and exclusion (Incl, Excl) criteria | Sample size (Age range; mean (SD)) | Sample characteristics (Sex (%); Ethnicity N(%)) | Summary of results |
|---|---|---|---|---|---|---|---|---|---|---|---|
| MacArthur (2020) | USA | To examine the risk and protective factors affecting medical students | Qualitative, phenomenology | Constructivist | Medical school campus | Reflective writing completed as part of wellness curriculum | Interpretive description approach | Unknown | 105 (Unknown; Unknown) | Unknown; Unknown | Depleting factors included stress and the structure of the curriculum, replenishing factors included hope for the future in interacting with patients |
| Van Rooji-Peiman (2020) | Holland | To gain insight into the role of psychological capital in the retention of first-year computer science students | Qualitative longitudinal case study, phenomenology | Constructivist | University campus | Individual interviews (three rounds) and graphic elicitation exercises** | Thematic analysis | Incl: First year computer science students selected as having variable previous qualifications and higher education experience, and variable prior programming experience | 16 (Unknown; Unknown) | Female 12.5%, Male 87.5%; Unknown | Hope was found to influence self-efficacy, resilience, and optimism |
| Wilson (2012) | Australia | To rebalance the individual and sociocultural dimensions of resilience and explore ways in which resilience is accumulated, preserved, and transferred | Ethnography, unknown | Constructivist | Community | Individual interviews | Unknown | Incl: 18–25-year-old Sudanese people living in Australia and people who support them | 10* (21–25; 23.0(1.05)) | Female 30%, Male 70%; Unknown | Hope was linked to educational aspirations, belief in God, roles models, and despair |

*Note:* Studies by Chamodraka, Hulme, Jones, Van Rooji-Peiman, & Wilson are theses identified through the grey literature searches. *Student subsample extracted. **Qualitative subsample and data extracted.

Steps were taken to try and promote credibility and reflexivity. NA and CB independently coded all data and used their respective familiarity to increase the credibility of the analysis. NA and CB both maintained an independent audit trail of codes and their evolution into themes throughout. NA initially led the analysis, due to CB's familiarity with the cognitive model of hope and the desire to ensure that this was not inaccurately prioritised above other conceptualisations. The thematic structure presented prioritises the patterns and interpretations identified by NA. The refinements made by CB reflected the simplification of the thematic structure, i.e., collapsing "thin" subthemes into themes for parsimony and coherence [30]. However, new themes (2,8,9) were additionally identified by CB. These two authors met to discuss the themes on multiple occasions. The third author provided a sense-check of the thematic structure and its correspondence to illustrative data presented.

## Quality appraisal

A quality appraisal of each reviewed study was performed by CB using the Mixed Methods Appraisal Tool [MMAT: 33]. The two screening questions were applied to all studies, then the relevant quality appraisal questions pertaining to the methodology (qualitative or mixed) were applied. Each question was rated as 'yes', 'no', or 'can't tell'. A summary of the quality appraisal is provided in Table 1, and the full rating schedule in supplementary materials (S1 Table).

In addition, the Confidence in the Evidence from Reviews of Qualitative Research (GRADE-CERqual) approach [34] was used by CB. This involved assessing the extent to which the findings from the review were a reasonable representation of the phenomenon of interest based on methodological limitations, coherence, adequacy, and relevance of data supporting each finding. The four domains were each assessed as no or very minor concerns, minor concerns, moderate concerns, or serious concerns. The overall assessment of confidence was rated as very low, low, moderate, or high. A summary of the GRADE-CERqual findings is provided in Table 3 and the full rating schedule in Table 2 of the supplementary materials.

**Table 3. Quality review summary and representation of themes across reviewed studies.**

| | Study quality appraisal summary (% criteria satisfied) | Hope is fundamental | Hope is the construal of self over time | Hope is goal-directed | Hope is thinking, doing, and feeling | Hope is connection | Hope is resilience | Hope is dynamic and reciprocal | Depression is the absence of hope | Hope is a positive, borne from a negative |
|---|---|---|---|---|---|---|---|---|---|---|
| Chamodraka (2008) | 100 | X | X | X | X | X | | X | X | X |
| De Pretto (2020) | 40 | | | X | X | X | | | | X |
| Hulme (1997) | 100 | X | X | X | X | X | X | X | X | X |
| Jones (2015) | 80 | X | X | X | X | X | X | X | | X |
| Lam (2017) | 20 | | | X | X | X | | X | | X |
| MacArthur (2020) | 60 | X | X | X | X | X | X | | | X |
| Van Rooji-Peiman (2020) | 100 | | X | X | X | | | X | X | X |
| Wilson (2012) | 60 | | | X | X | | X | | | X |
| Theme confidence appraisal summary | | | | | | | | | | |

## Results

### Study characteristics

At title-abstract stage, 487 papers were reviewed and 226 were sought for full-text review. Of the 214 available full-texts, eight papers were included in the final evidence synthesis (Fig 1). Study characteristics are summarised in Table 2. In total, three studies were conducted in the UK [35–37], two in the USA [38, 39], and one each from Canada [40], Australia [41], and China [42]. One study [35] focused on the experience of hope among students. Six studies [36–41] examined hope in relationship to other concepts such as optimism, resilience, and despair. The remaining study [42] focussed on the role of hope in academic failure and success.

Six studies [35, 37–40, 42] focused solely on the experiences of higher education students. Two studies [36, 41] focused on young adults in general, although only higher education student data were extracted. Of the higher education student (sub)samples, one study involved a general sample from private universities [43]. Three studies involved students studying a particular discipline (psychology [35]; computer science [37]; medicine [39]). Four studies involved students defined by other characteristics that were considered to be especially relevant to the experience of hope. These characteristics were: receiving psychotherapy and

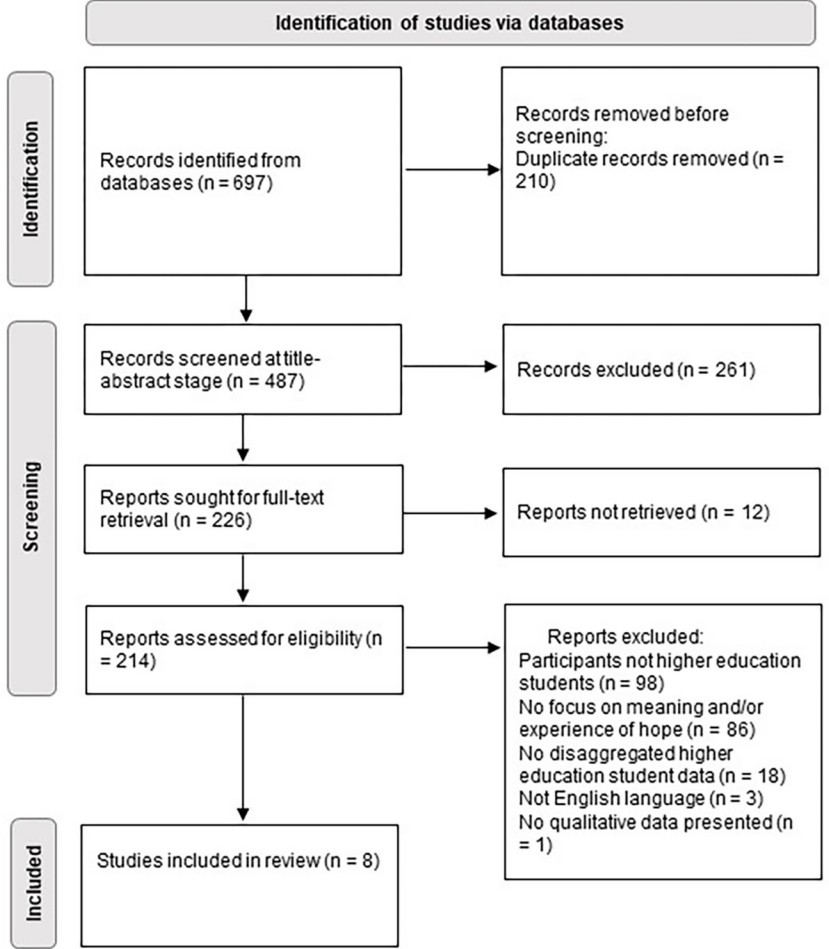

**Fig 1. PRISMA flow diagram showing literature screening stages.**

evidencing a gain in hope [40]; being hopeful despite educational and family challenges [42]; being a looked after child [36] or migrant [41]. Sub/samples ranged from 4–105 higher education students. Studies generally reported the age and sex or gender of their participants. The age of included participants ranged from 18 to 52 years, where known, with an average of early twenties. Most studies involved both female and male participants. Only two studies reported participants' ethnicity [36, 40].

The majority of studies took a constructivist epistemic stance. Seven studies were qualitative, and one [42] was mixed methods. Six studies collected data using individual interviews, of which one study additionally used group interviews and collected diary entries [43]. The two remaining studies used qualitative survey data [35] and writing extracts respectively [39]. All papers were reported in the English language.

## Quality appraisal

The quality of each individual study was assessed using the MMAT tool [33]. All studies were identified as of high quality with respect to the two screening items. The qualitative studies (n = 7) were of variable quality, ranging from satisfying 40% to 100% of the relevant criteria. Three studies [37, 40, 43] were scored as high-quality relative to all criteria (i.e. 100% "Yes"). One study [36] satisfied four (80%) of the elements, but did not sufficiently substantiate the results with data, as in-depth interpretations were made at times with no textual evidence. Two studies [39, 41] satisfied three (60%) of the elements but did not collect adequate data [39], fully substantiate their findings with data [41], or adequately derive findings from the data [39, 41]. The final qualitative study [35] satisfied 40% of the criteria. It did not collect adequate data, adequately derive findings from the data, or fully substantiate their findings with data. Thus, the higher-quality elements of the studies overall were that a qualitative approach was appropriately used and there was coherence between the data sources, collection, analysis, and interpretation. The elements that were more commonly lacking were the articulation of the analysis method and the substantiation of the findings with sufficient textual evidence. The studies that collected more in-depth qualitative data using interviews, tended to score more highly than those that collected briefer excerpts from writing examples of qualitative surveys only. The single mixed methods study [42] failed to satisfy 80% of the criteria. The quantitative and qualitative aspects of the study were well-completed in isolation. However, there was a failure to justify the mixed methods approach, to integrate their findings, interpret the integration, nor to examine convergences and divergences across these components.

## Evidence synthesis

Nine themes were identified. Each is described in turn, with illustrative quotations presented with their source. An account of the representation of each theme within the reviewed studies is provided in Table 3.

## Theme 1: Hope is fundamental

This theme positions hope as fundamental, something of immense importance and an inherent part of the human experience. Students spoke of hope as an inevitable certainty, something ubiquitous and "innate" [43, p98]. Hope was considered a fundamental characteristic of being human; "It is a characteristic of the soul" [43, p91], "It is inherent in the human composition" [43, p121]. Therefore, hope was described as something that everyone has, albeit with some variability in the amount; "Some people have been given more than others, but it is present in some form in every person" [43, p91]. Nonetheless, hope was so powerful that even a small amount could be incredibly impactful:

. . .just the glimmer of hope changed my life and made me do the actions of getting out [43, p101]

Even the slightest glimpses of the future ignites my fire [39, p6].

Therefore, hope was considered fundamental, although variable, yet with even a small amount having an effect.

For some students, hope was built upon a sense of connecting with fundamental existential meaning, specifically with respect to belief in God or another spiritual being; "putting God as part of my life [is] the only thing give me hope" [41, p99]. For religious or spiritual students, it was argued that it was not possible to separate faith and hope; "I believe that hope is an integral part of faith, and that faith is an integral part of hope. Indeed, one cannot exist without the other." [43, p86]. Specifically, students suggested that hope was linked to faith in there being a plan for their future, which inspires positive action:

People who believe in something, whether it be a god or goddess, or a nature spirit tend to be more hopeful for the future. People who have faith that there is a master plan are able to believe that they have a good place in it and often strive harder to fulfil their role. [43, p88]

The previous quote additionally suggests that hope linked to faith may be conducive to motivated goal pursuit.

The conceptualisation of hope as fundamental was imbued with a sense of positive determinism; ". . .it is the one-word definition of looking forward to tomorrow because it will be better" [43, p84]. The use of "will" in the previous quote, and "expectation" and "reality" in the following, reflect the sense of certainty that the desired future will occur; "Expectation, something you are looking forward to, it's like a reality coming to pass" [36, p106]. Nature metaphors were one way in which students positioned hope as something innate and intuitive; "Hope is confidence that the sun will rise again tomorrow" [43, p84]. Similarly, a comparison was made between hope and breathing, additionally emphasising the critical necessity of hope; "it is like breathing, one must have it to live. Without it one will certainly die." [43, p121]. The use of metaphor was helpful for students in articulating the nature of hope, which they found to be so intuitive, it became almost impossible to define:

I am sorry, it is difficult and nearly impossible for me to put a deep feeling into a properly worded definition [43, p92]

Hope is intangible, impossible to define [43, p121].

Despite its importance, this conceptualisation of hope was inherently passive, i.e., hoping appeared to be something that just 'was', rather than something created or maintained by choice or effortful action. The following quote reflects that this conceptualisation of hope involves a more external locus of control, i.e., positioning hope as more a property of the world than a property of self; "Hope is placed in something that will come to be or has the chance to be" [43, p85]. Similarly, the certainty of the positive future occurring did not appear contingent on any actions taken. Within this definition, little to no distinction was made between hope and optimism, ". . .they are barely different, they are so close" [43, p122], because they were so similar.

### Theme 2: Hope is the construal of self over time

This theme described how hope scaffolds the construal of oneself over time, i.e., it is used as a process of connecting past and present experiences with who one desires to become in the

future; "[Hope] is something that affects the very core of who you are" [43, p176]. Hope is a way of creating meaning, by making sense of past experiences and using the hoped-for future to motivate behavioural change in the present moment.

Hope first involves the imagination of an ideal future self, i.e., the identification of the type of person the student wants to be in the future; ". . . it's about where you finish the race" [41, p179]. This imagined future self was typically described as constituting visual imagery, albeit with varying clarity and vividness:

> I kind of do have pictures. I can't picture what my kids look like or my wife. But I do have kind of a picture of a nice house or backyard or just sitting on the porch with my wife and watching them play. I can visualize that [43, p155].

The following extract describes how this hoped-for self-image provides motivation to keep working towards specific goals and to engage in facilitative behaviours in the present moment:

> I think I always have had in the back of my mind that I have always had an image of myself that I was kinda like an 'A' student and I think that helped perpetuate it because I've had friends who thought of themselves as 'C' students or always saying I won't do well on the test, this class is awful, I can't understand it, and it seems like they would always be getting lower grades and so I always tried to keep up that image of myself that I was a good student, because then I felt more, I just felt smarter. If I kept to the image, then I felt like I could handle it. [43, p93].

This extract additionally identifies the risks of focusing on a less-than-ideal future self, which may create a self-fulfilling prophecy in which this not ideal self then comes to be.

The imagined future self-image functions as a totem, providing a sense of meaning and purpose to one's existence over time; "Hope allows you to find meaning in your life, in your work, in your reason for going on". [43, pp91-92]. Through providing this sense of meaning, the hoped-for self validates one's sense of personhood:

> I had a sense of value, and purpose and direction that for me was hope that there was meaning and the hope was in spite of my present reality. That again there was a value and a reason for being. And that helped to dissipate the sense of directionlessness and provide or sort of validated the fact that I existed. [43, p180].

Therefore, the ideal future self provides a bridge between different self-states across time; ""I think hope. . .is past, present, and future. Knowing what you have been through, knowing what you are going through now, and knowing that in the end everything is going to turn out" [43, p186]. This bridge is one of self-understanding, making sense of connections between past experiences and present behaviour, and what behavioural change is needed to work towards the desired future; ". . .there is a very direct link between understanding myself better and being more hopeful" [40, p101]. The bridge not only functions as a link, but also a route by which to escape feeling entrapped by the past or present:

> That's one of the big wolf behind me, in my mind. Every time I get down and feel like I'm gonna give up- that wolf behind me, which is the ghetto where I came from, is be running after me even though I'm trying to get away from it because I don't wanna go back there [36, p145]

> Without hope you get caught in the finality of the present [43, p179].

This escape was necessary because the (hoped for) future self was considered to be more important than the past or present selves, because it was more desirable; "It's not where you start in life that matters. . .I think it's the finish that matters" [41, p179]. Nonetheless, for some students, especially those who were more vulnerable, a far-reaching future self-image seemed overwhelming:

At the moment I reframed my hopes for the future to next week as opposed to three years from now. So yeah I think I probably, as a result of the counselling experience (incomprehensible) sort of being happy and being able to cope with the immediate future more than succeeding at everything you want to achieve. [40, p71].

Therefore, it was important for students to have a sense of an aspirational future self that did not feel too distal. A more immediate future self-image felt more motivating because it felt more achievable.

## Theme 3: Hope is goal-directed

This theme identified hope as inherently goal-directed, with goals linked to the pursuit of one's hoped-for future self. The mental invocation of a desired goal becomes the focus of hope and, the more specific and realistic the goal, the more it occupies and sustains one's hopeful thinking.

A specific goal was identified as the object of hope, that which became the focus of students' hopeful thinking and associated emotions and behaviours; "With hope there is a focus, there is a goal. . . in other words there is a sense of direction" [43, p123]. Spending time thinking about an important goal could enhance students' level of hope; "It's almost like my goals and everything, my aspirations tied together, they give me this hope that things will be different" [36, p108]. To be meaningful, and therefore to enhance hope, the goal needed to reflect the student's construal of a positive future; "The thought of the future, caring for people as a physician, drives me every day. . .Even the slightest glimpse of the future ignites my fire." [39, p6]. Students could then derive a sense of motivation from thinking about a meaningful goal:

Because of the hope, you will move forward, keep moving forward (on the road) and complete what you want to complete [42, p139]

Hope is motivation. If you do not have hope, you will sink (to the bottom), and you will not reach the goal. You will find the goal is far from you and then you will not pursue it. But if it is hopeful to attain the goal, you will be motivated to pursue the goal. [42, p139].

Examples of identified goals were typically oriented around work and education, family, relationships, and lifestyle, for example; "I have this hope because I want to study and become a child psychologist in the future" [35, p5499]. Goals were nonetheless emphasised as something unique to each individual; "Hope is built on the individual road" [42, p139]. Thus, goals are the unique and meaningful building blocks upon which hope is focused.

A goal alone was not sufficient, however; both a specific goal and a sense of hope were deemed necessary to ensure the student would experience success:

If you have hope and strongly believe in what you're trying to learn you will work harder at what you are learning, you will tend to give it your all. If you don't have much hope in whatever you're doing than you're not excited about it, you're not willing to give it the best you've got. If you don't have hope and a goal, then it becomes pointless and you'd tend to

give up easily. But on the other hand, if you've got a goal and have hope, then you tend to achieve much higher, and you often reach your goal with enthusiasm. [43, p144].

Moreover, there were two necessary conditions to ensure that goals occupied the focus of hope and did not undermine it. First, goals must be specific and second, goals must be realistically attainable; ". . .the clearer, more realistic I think your goals are, it's easier to maintain your hope" [43, p158]. The importance of goal specificity is articulated in the following extract, which describes how the more specific a goal is, the easier it is to visualise it and develop a sense of emotional connection with the desired outcome; ". . .the clearer the picture is, the more you can get excited about it and hopeful" [43, p167]. It is this specificity of hope in relation to one or more goal within this conceptualisation that offers some distinction from optimism; "Hope is more that in this particular case or in this specific situation I believe it will work out for me. [. . .] And optimism is more: we can do this, this is going to be all right" [37, p101]. Optimism is contrast was identified as a vague sense that things would be okay.

For hope to also be sustainable, goals must be achievable, i.e., one must believe their goal is reachable, and be able to identify the specific steps to doing so; ". . .it's easier to maintain your hope as you see yourself reaching the goal step by step" [43, p158]. Without the belief that the objects of hope could be realised, hope would be much undermined: "And if it is not attainable anymore, then your hope diminishes like: this is not going to work anymore" [37, p116]. Students must also observe that the logistical conditions are in place for them to be able to actually reach their goals. Necessary logistical conditions included being confident in one's competence, ". . .that I consider myself competent enough is probably a greater contributor to my hopefulness" [43, p92], and in the availability of opportunities and resources for necessary for goal pursuit:

I have no academic qualification, which means I do not have hope. If I cannot get it, it means I have no hope [42, p138].

I failed (the public exam). My family asked me to stop my study and find a job for living. . .We are not rich. The only way (for further study) is the financial aid from the government'. [42, p138].

The experience of failing at something connected to a specific goal could act to undermine hope, by making the goal seem unachievable:

When I getting a bad grade in school. Like for instance, my maths, I'm not really good at maths. We just had a mid-term exam and I know I didn't pass it, so I know that right there that causes me, that deters me from being hopeful for the future." [36, p134].

However, bringing one's focus back to the broader goal could resurrect one's sense of hope; "You have to think about your goal. You have to know your way. I don't want to give up just because of the (academic) failure" [42, p139]. A meaningful goal could inspire motivation despite logistical setbacks because the goal functioned as a step on the journey to a desirable and purposeful future.

## Theme 4: Hope is thinking, doing, and feeling

This theme reflects the conceptualisation of hope as cognitive, behavioural, and affective. Students did identify these components variably. Some students identified hope as primarily cognitive or affective, whereas some emphasised the interplay of these different facets.

First, the description of hope as cognitive emphasised the notion of hope as an optimistic, self-agentic thinking style; "I had some power that I felt that I had a greater sense of being in control. You know like 'it's okay, I can manage this'" [40, p66]. The act of adopting a hopeful thinking style was itself agentic. Students identified needing to make a conscious choice to view things positively and not undermine their own sense of hope; "…it is a choice because I've had some painful times recently and I had to decide, 'Am I going to be hopeful about the things that are positive or think about how things are going to go bad?'" [43, p119]. It was possible to preserve one's hope even in difficult circumstances, through the use of positive self-talk; "…because if you think: I can't do this, the others [factors] will go down. And if you think: I can do this, then your hope increases and the others follow" [37, p116]. The deliberate adoption of a self-agentic thinking style, in turn, scaffolded a movement from a more passive optimistic stance to planning for the active pursuit of a positive future; "…you don't just hope that something will be okay, you start to have confidence that it will and that there is a way out and that there is some stuff you can do" [40, p68]. Inversely, students described how it was also possible to choose not to be hopeful; "…you could choose not to be hopeful just by your attitude. I think, generally, I would have to say it is a choice, because I know some people who choose not to be hopeful" [43, p119]. It was emphasised, however, that it is typically a gradual iterative process by which one adopts a non-hopeful mindset: "I don't think anyone says, 'Boy, wouldn't it be better to be despairing than hopeful?' I think it is a series of choices that lead to that, that create that ultimate choice." [43, p120]. Therefore, the adoption of a pessimistic or low-hope mindset does reflect the exertion of self-agency but not necessarily in a wholly conscious or considered way.

The other identified components of hope were behaviour and emotion. The act of 'doing' was linked to experiencing hope: "So this [the music project] is like, a good opportunity for us tell the youth. We have something else to do, instead of just doing nothing" [41, p241]. Specifically, the behavioural conceptualisation of hope focused on performing meaningful activities that were linked to an overarching sense of purpose; "I think for me hope is the sense that there is value in the things that I do and there will be value in continuing to do them—purpose" [43, p180]. Other students identified hope as emotional in nature; "I do think it is a feeling so I definitely think that it's an emotion" [43, p118]. This description typically involved describing hope as a positive emotion. However, hope was also conceptualised as the absence of negative feelings; "…a peace of mind, Everything is gonna be ok, you don't have to worry about anything" [36, p106]. Therefore, hope was described as meaningful action or as emotion that either was positive or involved no negative affect.

Nonetheless, even students who described hope in affective terms emphasised that this was not sufficient, that hope was something more than just emotional; "It's not really a feeling as much as it is a state of… well not mind, but perhaps a state of being. It's that voice inside of you that tells you things are going to get better [43, p97]. The effects of hope were notably broad and widespread: "I think it is a deep feeling because once you make that, it affects everything. I mean, it affects your plans, your level of happiness, I feel like." [43, p121]. Thus finally, some students identified hope as an interplay between cognitive, affective, and behavioural facets:

It is a thought… with actions [42, p139]

…to get a job, I just need to learn. I find it as an opportunity, but when I was in Africa I would have not done it…Yeah, so it has been in my hopes to study. So, I'm so happy [41, p121].

This was hierarchical, in that cognition was typically identified as the most important component, "There is definite interplay between emotions and choices, but I believe that it is ultimately the choices that you make that will lead you there" [43, p120]. Cognition was deemed to activate emotions and behaviours; "you could have like a happiness inside you when you think of what you expect" [36, p109]. Yet emotion was emphasised as needed to really drive goal-directed behaviour:

> . . .if you choose to hope, but you don't have the feeling there, you are not really going to be able to do it, but you also have to make a choice to let that emotion lead you if you really want to be hopeful [43, p121]

> "I think it's the idea of being able to be happy for a bit of time. Which I have no idea if that's how I defined it before but it probably had something to do with accomplishing stuff. So, accomplishments are good but being happy is also important [40, p71].

Moreover, these examples demonstrate that there would be a kind of emptiness to the pursuit and achievement of goals without emotional responses–as well as this absence likely undermining success.

## Theme 5: Hope is connection

This conceptualisation identifies hope as being built upon relational foundations. The influence of others on students' hope included both those with whom they had intimate relationships and it occurred outside of close personal relationships too.

Hope was described as arising in the context of meaningful relationships; "It can be built up by the knowledge that others support and love you" [43, p176]. The influence of other people was not necessarily predicated upon there being a close personal relationship. It was possible for hope to be enhanced too in less proximal relationships; especially those in which the other evoked a clear association with past or future self-construals. In fact, hope might be best enhanced in the context of less close relationships. This was perhaps because encouragement from other people was seen to be more objective and thus more meaningful; "It is always encouraging when someone believes in you other than your parents" [43, p101]. Moreover, it was relevant as to whether the other person was in a position of authority:

> The first year I had renewals, [faculty member] Bill Jones said to me, I was considering options and he goes, 'you should really consider being a college level professor. It's so much more academic freedom.' And I remember him saying that. And that is the only experience I have ever had with Bill Jones. I don't think he'd know who I am. [43, p102].

Being noticed and encouraged by an authority figure appeared to have a powerful impact on hope.

There were multiple mechanisms identified as to the means by which relationships impacted hope. First, students described how other people could act as role models of hope for them:

> Basically he [mentor] has similar circumstances like myself and overcame all odds. Got honoured by the Queen of England. He has a charity for orphans and donates a lot. I give him credit because he's always been there when I was down to encourage me to keep on pushing [36, p129]

My adopted dad had a lot of obstacles himself as a young teenager as well growing up. And he didn't have it easy and people think he had it easy. I always tell him, you are my role model and when I grow up, I want to be like you. And he's like a really big push in me doing well and me wanting to be like him, or be a version of him. [36, p125].

Thus, students could use role models as evidence that they could reach their desired future and examples of perhaps how to do so.

Second, the presence of someone else who identified that the student was struggling and 'reached in' to offer them help:

. . .when you are in that state of being you feel like you're in the dark and hopeless, it is important that someone comes along beside you. . . That he, in the dark, kind of takes you and shows you the way to the light and sometimes when you are in that hopeless state of being you don't always see that there is a way out or you don't see that there is anything to be hopeful for [43, pp97-98].

It appeared important that the individual offering help did so from a position of understanding:

It was just the fact that someone understood what I said that I needed something specific. . . they sort of really made the effort to help me with what I said I needed. . .that I really appreciated and that certainly gave me—that certainly gave me I guess hope. . .she really did cater to what I needed and is really there to help. [40, p108].

In this way, other people could use their prior shared experience to increase students' sense of mattering and their motivation to keep pursuing their goals during difficult times.

Third, other people could actively encourage self-agency and engagement with one's goals. The first example describes a mentor encouraging the student to stay connected to their desire for their goal and engage in behaviour likely to contribute to its achievement. The second example describes a friend helping the students to spot their own strengths and challenge negative self-construals:

. . .you trying to do good in school, but then you want play with the rest of the children, and neglect your studies or neglect what you had to do. She would push you back, "You say you want to do this." She's one of the persons in my life now that is still steering me to move forward. She give me words of encouragement like like "Go girl!" or "I know you always had it in you!" or stuff like that. [36, p128]

My friend listened to me as I put myself down and gave 100 reasons why I should NOT get an internship. . .She sat with me for an hour praising my writing and encouraging me. . .Then she had me list all my positive qualities and reasons why I wanted the internship. Suddenly I began to realize how important this internship was to me and I was back on track. [43, p101].

These examples demonstrate how encouragement is an important part of both professional or formal and personal relationships.

Finally, students identified mutuality as a source of hope. The following extract situates hope within the context of both a sense of being truly known by other people, in this case peers in the student cohort, as well as receiving explicit encouragement. The use of 'we' implies a sense of mutuality, i.e., that hope comes from giving as well as receiving encouragement:

I have made friends that have become my family. They know me inside and out. They know how I think. They know how I act. What food I love and hate. What my habits are. They have seen me at my best and at my worst, and have always cheered me on at my best and at my worst. They have almost become my siblings. We bicker, fight, laugh, and push each other to do our best. [39, p7].

Similarly, students described wanting to serve as a source of hope for other people, inspiring them through their own success; "My hope is that I could help the kids of the orphanage. My hope is really to shed light on orphans worldwide." [36, p146]. The influence of mutuality could be however a double-edged sword. Whilst students really wanted to help others, doing so might involve re-experiencing trauma, either through revisiting specific places or seeing their peers encountering familiar difficulties:

I know a lot of people that grow up with me, a lot of girls, a lot of guys, are pregnant, some are in jail, sad to say, two died. It really hurts me because here it is I was the model child, academically inclined, everyone looking at me to help them out [36, p141]

I would like to go back and help the people who in the ghetto, but I don't wanna be there. . . some people will look at us today and say they never even grow up in the ghetto. But we lived it, breathe it and I don't want no kids like I say in the ghetto to think that they don't have no future for themselves. . .they don't even know the story. They don't know that I was once where you were or even worse. . .And one of the things I wanna do too is when I get my degree and when I finally have enough money to do certain things I wanna do like an after school programme in the area where I used to live. [36, p146].

Thus, whilst hope-enhancing, mutuality was additionally one way in which the relational elements of hope could be challenging.

There were two other examples of interpersonal factors undermining hope. First, a perceived lack of social connectedness and support was deemed to undermine hope; ". . .it is hard to be hopeful about the future when you don't know from where you are going to be getting your love and support" [43, p99]. Second, hope could be undermined by other people being pessimistic and discouraging, or conversely, having too high expectations of students:

They (parents) don't consider my situation. . . They cannot see that there is a hope for me. I am not like a university student in their eyes. They look down on me. They think I am hopeless in study and I should stop my study [42, p138]

They were always encouraging me to do things, like music, to keep my grades high and they would reward me. Actually they rewarded me at first and then it just became expected. It got more and more. . .Perfection is not always possible. [43, p140].

Nonetheless, it was clear that there were individual differences in the risk of interpersonal circumstances being able to undermine hope:

Success does come with a lot of backlash. That's what I'm facing even as I move further in my life. . .even though people don't like me because I am doing well. It is very puzzling to me. I just use it to push me [36, p142].

The above excerpt shows how, in the context of high hope, an interpersonal 'backlash' from success could be used as a form of motivation to keep pursuing one's goals.

## Theme 6: Hope is resilience

This theme describes how hope confers a sense of resilience, helping students to cope with challenging circumstances and protecting them from the negative impacts of difficulties. Students emphasised how hope is perhaps most keenly felt in the context of adversity; "Hope is what rises up and grips the soul in the darkest night, inspiring it to reach beyond the darkness" [43, p97]. Challenging experiences were emphasised as being a breeding ground for a more authentic type of hope:

> [Interviewer: Do you have a different perspective on hope since you have been through so much pain?] I think it changes your perspective on life so it changes your perspective on hope. You, instead of just being like, 'oh I hope I get a good job in the future.' You really begin to live hope. It's not something I thought about a lot 'oh wow I have hope' but you really begin to live it. [43, p157].

Adversity thus seemed to provoke students to very engage actively with their sense of hopefulness.

The experience of hope in adverse circumstances provided motivation to believe that positive change is possible; "It's that substance that makes you persevere when the world is caving in, I think hope is that innate ability to see that little path or see that little glimpse of light, or a bridge, and then follow that" [43, p98]. As such, hope acts as a buffer to protect students against the negative impacts of adversity and trauma. In the following extracts, students highlight a transcendent quality to hope. Hope affords students the ability to see beyond their past or current challenges, to position these challenges as small parts of a larger life story, and to identify new solutions to their problems:

> [Without hope] we are caught in other problems and we've made ourselves a very—narrow—view of what exactly it is that is wrong and maybe sometimes this is what hinders us to find solutions because we're just–so much obsessed (her emphasis) with what is wrong that we don't see how we could actually, attack it from a different perspective. [40, pp85-86]

> And so if you're having a bad day, that's OK because that hope transcends that bad day. But without any hope, you're caught in whatever that moment brings and you don't see beyond it You don't have a perspective. You don't see the whole picture—it's like you're caught right there [43, p153].

Students emphasised that thinking specifically about the goals they wanted to pursue helped them to stay motivated during difficult times; "[Thinking about my goal] fuels me through the frustrations and challenges" [39, p6]. Hope could also buffer the negative impact of not achieving something desired or experiencing barriers on route to a meaningful goal:

> It's that voice inside of you that tells you things are going to be better for you soon—even when outwardly someone else couldn't even stretch to find any good coming to your less-than-ideal situation. You can hope even when you are in the middle of a totally demoralizing situation—when everything is against you and there's no light at the end of the tunnel. [43, p97-98].

These excerpts demonstrate how hope enables students to find meaning even in the most negative circumstances, and that when all else fails, hope remains.

### Theme 7: Hope is dynamic and reciprocal

Students spoke of hope as dynamic and reciprocal. They emphasised that hope waxes and wanes over time and in response to different life circumstances, but also that the components of hope influence each other in non-recursive feedback loops.

Students first identified that hope is not static, but rather changes over time, "My level of hope fluctuates" [43, p114], and is influenced by life events and experiences; "I can say that I'm hopeful and I'm going to do all this stuff but I really haven't faced anything that's too challenging or that's made me have to take a completely different look at life" [43, p131]. Students described how the different components of hope influence each other:

> I mean like once the grade report comes out as a result of my effort, it makes me feel like I can do it. Yea. It [good grades] makes me feel like I still could become what I want to become [36, p133].

> I was feeling more hopeful so therefore I started myself to make efforts to apply what we had been talking about and just being more positive. You know, in my everyday life. So it's a circle. It's some kind of a chain of because I became more hopeful, I started you know making efforts and thinking more positively, and then because I was thinking more positively, 'yeah you can do it, you'll be able to think that way more often' [40, p114].

The preceding quotes describe how hope manifests as a cycle of hoping for specific goals, engaging in hopeful thinking and behaviour which raise motivation to then engage in goal-directed behaviour, observing positive impacts of these behaviours, and then experiencing even greater hope. Students commonly used analogies of circles or chains to express this non-recursive nature of hope.

The next excerpts describe moreover how other positive cognitions and emotions (like confidence, resilience, and optimism) and hope reciprocally influence each other:

> It is like a circle, it influences each other. If you have hope, then optimism rises and if you have optimism then you sort of develop expectations of yourself like: I'm doing things on my level now, I can do this. When you receive a bad or a good grade then resilience kicks in like: oh it went really bad or this went really well, but it goes around in a circle, sort of. They are separate from each other, but they are connected to each other." [37, p116]

> "I think they build on each other, but I think confidence probably helps to build your future hope and your day to day hopes probably help to build your confidence- So they are building on each other…" [43, pp92-93].

Thus, hope is reciprocal with respect to both its constituent parts working together and in terms of influencing, and being influenced by, related cognitions and emotions. Together these influences contribute to hope as self-reinforcing and able to grow over time and experience.

### Theme 8: Depression is the absence of hope

This theme reflects the identification of depression as the total absence of hope, and that students experienced depression as intertwined with a sense of a lost future.

Students described the experience of depression as comprising a sense of hopelessness, as opposed to hopefulness; "…you could almost parallel it with total depression, maybe. I don't know, just total, absolutely no hope" [43, p105]. If causal, the relationship was described as that a loss of hope for a positive future influenced the onset of depression:

Dashed hope sends one into depression [43, p105]

. . .my dream was to play in a professional orchestra 'cause that's my favorite thing in the world. . .That was another part that led to the episode in the hospital. My dreams were falling apart [43, p115].

The mechanism appeared to be *via* negative cognition, i.e., that depression arose as an outcome of pessimistic thoughts about oneself and the future. Students might still have goals. Yet students lacked a sense of hope that they could achieve them, perhaps because the goals themselves were not necessarily ever really achievable; "It was like if I don't get A's in everything, then, you know, I have failed. That was the kind of thinking that led to the problem with depression." [43, p140]. Nonetheless, students suggested that they still had self-agency in this type of scenario. They emphasised that everyone can exercise some choice about re-connecting with a more hopeful thinking style:

You could decide to go for something else and then get hope again or go into depression until you get around to something else [43, p105]

About a year ago I had to make a decision to either focus on the negative things that have happened to me or to focus on the positive. For most of my time in college I focused on the negative. As a result I became increasingly depressed until I was no longer able to face the trials of every day life. [43, p105].

Therefore, for example, students emphasised that people could choose to select a different, perhaps more achievable goal, or alternatively to choose to retain a pessimistic stance that would serve to maintain depression.

If experiencing depression, the regaining of hope was connected to the realisation of one's potential and the ability to envisage a meaningful future, even if needing to relinquish previously valued goals that may no longer seem possible:

I ended up temporarily dropping out of school and checking myself into a psychiatric hospital. This was the most difficult thing I have ever done in my life, but I believe it was necessary. If I had not done something to stop me in my downward slide, I am afraid to think what might have happened to me. While I was in the hospital I realized that despite the fact that my dream of being a professional musician was dead, I still had a lot of promise [43, pp105-106]

Hopelessness can come from inaction, just not being able to do what's normal for you. You know that can lead to depression and hopelessness. [43, p107].

The latter quote demonstrates too that a loss of sense of functional capacity in the present moment, as well as loss of future aspiration, is linked to a lack of hope.

## Cross-cutting Theme 9: Hope is a positive, perhaps borne from a negative

Across all themes the description of hope was nearly wholly positive. Hope was focused on a positive and desired future and the identification of positive goals. It involves a positive thinking style, with cognitions that are optimistic about oneself, other people, and the future; "Hope is the ability to see the positive side of a situation or person" [43, p176]. Hope was also associated with the experience of positive emotions and the ability to cope adaptively with challenges. As demonstrated in the following excerpt, the positive nature of hope was well

articulated through metaphorical descriptions of it as being a source of "light"; "It's like as if I'm in a completely dark room that's without any guidelines and all of a sudden there is a little–in dark theatres sometimes they have these tiny little light-bulbs that help you find your way." [40, p68]. This figurative language evokes connotations of hope as reflecting love, comfort, understanding, connection, and salvation.

Nonetheless, hope was often described as being borne from something negative. Hope involved students observing or experiencing something aversive, either something expected that was lacking or something traumatic. This was coupled with the recognition that the student did not want to re-experience this aversive experience, but rather desired a different future:

> I focus on being a better person and that actually motivates me because, I say to myself "I don't wanna be like my parents", or "I don't wanna be like my dad." What he did and left and didn't take care of his child. And I look at it and say I don't wanna be like that. [36, p144].

Hope was enacted in this space of liminality, providing a comforting sense of motivation and the possibility of movement away from past and present uncertainty and adversity; ". . .we sometimes feel like, if you are in a very tough time, you are near something good" [41, p100]. As such, hope was experienced as a kind of "longing" [36], identifying itself in a gap that students desired to fill:

> Hope is also the feeling left when all other feeling is gone. One can hope for love, happiness, security, etc., even when all those feelings are gone. Hope is generally a feeling for something better [43, pp118-119]

> The desire to have your own family derives from the fact that I experienced several shortcomings and I hope to fill them with my future family [35, p5499].

Therefore, whilst hope seemed to commonly make itself known in the context of adversity and loss, it was nonetheless a bridge to a more positive future.

## Discussion

This review identified and synthesised qualitative data on higher education students' conceptualisations of hope. The main findings of this review were that hope is a complex construct that can be difficult to describe but is nonetheless of great importance and salience within the life experiences of higher education students. Hope was conceptualised in multiple different ways: fundamental; integral to the construal of self over time; goal-directed; cognitive-affective-behavioural; routed in connection; conferring resilience. In all its guises, hope was positioned as positive in both nature and impacts. Nonetheless, hope's creation was commonly described as arising from negative experiences. In contrast, depression was described as the absence of hope.

There are evident connections between these different conceptualisations. The description of hope as an innate and central part of being human underlies hope's role in the construction of self over time. The dynamic reciprocity of hope reflects the interactions between the three constituent ways of experiencing it: cognitive; behavioural; affective. The conceptualisation of hope as resilience is predicated on hope helping students to cope with adversity due to their clarity of self-construal, their ability to identify meaningful goals and use self-agency, and their sense of connectedness with other people and/or spiritual being(s). There are additionally

some evident tensions. Although the different conceptualisations were largely not mutually exclusive, there is some tension between the conceptualisation of hope as innate *versus* describing hope as a self-agentic choice. Equally, the innateness of hope is somewhat at odds with the positioning of hope as relationally driven and affected by life events and experiences. The description of hope as innate was espoused mainly by students from a Christian university and thus may reflect a view of hope viewed through a lens of religiosity. Nonetheless, an innate capacity for hope does not preclude that hope may wax and wane over time or be affected by different experiences, with students then able to choose to actively engage with and increase their sense of hopefulness.

The GRADE CER-qual review resulted in the identification of two themes as reflecting conclusions in which a high level of confidence may be placed. These were hope as goal-directed and hope as a positive, perhaps borne from a negative. All other themes were considered to be conclusions in which a moderate level of confidence may be placed. Four themes were undermined by minor or moderate concerns regarding the adequacy of the data. Not all of the studies were focused on hope as their defining or primary aim. Thus, the amount of data pertaining to the conceptualisation of hope, and especially to its relationship with mental health and wellbeing, was limited. Moreover, there were methodological limitations pertaining to the individual studies reviewed which affected the assessment of confidence. Nonetheless, most themes represented most or all of the included studies, therefore, offsetting the limitations of each. Overall, the assessment of confidence in the conclusions of this review is that the key themes are well-supported by the underlying studies and are coherent and relevant to higher education students across geographical setting and subject discipline.

The findings of the current review are clearly in alignment with the cognitive model of hope [1]. Higher education students appeared to intuitively construct hope as a positive, goal-directed, future-oriented cognition. Like Snyder's model [1], students positioned hope as manifesting in resilience and as arising in the context of interpersonal connection. Moreover, students described how hope is dynamic and self-reinforcing. This aligns with cognitive hope theory which posits both feedback and feed-forward loops in that emotional reactions follow cognitive appraisals of having reached (or not) specific goals [1]. In turn then these emotional reactions influence self-agency and pathways thinking more generally, contributing to greater (or lesser) trait hope [1]. Current findings thus support the applicability of conclusions from cognitively oriented hope research to higher education students. This is positive, because evidence in the educational setting suggests that students' hope robustly predicts academic and vocational outcomes [26, 44, 45], over and above the variance predicted by IQ and ability [46]. Moreover, hope can be significantly increased in educational settings with even brief intervention, including that delivered by non-specialists [26].

Nonetheless, current findings do emphasise additional components to conceptualisations of hope. Namely, hope as fundamental and innate, hope as emotional and behavioural, hope as the construal of self, and hope as borne of negative experiences. The fundamentality of hope aligns with essentially all models. Whether in the context of psychology or philosophy and beyond, hope is described as of central importance to human experience [47]. The described innateness of hope, however, appeared to be more narrowly linked to hope as a more general and passive belief in a positive and meaningful future that arises in the context of religious faith. The broader body of students appeared to experience hope as more within their control. This aligns to the Snyderian model of hope as goal-directed willpower and waypower [48], i.e., hope as about one's ability to set and pursue specific goals. In addition, however, there was evidence of students conceptualising hope as emotional in nature. This could be argued to present a departure from Snyder's model [1], which suggests that hope does not comprise emotion. Nonetheless, it fits with Snyder's position that hope and emotion influence each other.

Specifically, for example, people who are high in hope arguably pursue their goals with more excitement, energy, inspiration, creativity, and enthusiasm [48]. Current findings, therefore, align with educational theory and research that suggests that hope motivates higher education students to be cognitively, emotionally, and behaviourally engaged in secondary and higher education [49]. In turn, this encourages students to perform adaptive and success-oriented behaviours that bring about positive academic outcomes [49].

The notion of hope arising from negative experiences aligns with Snyder's perspective that past experiences influence hope [1]. However, Snyder states that hope, especially self-agency, is contingent upon past experiences of setting and achieving goals [48]. Thus, past negative experiences in which it was not possible to meet goals should likely undermine hope. Nonetheless, the more individuals have had to overcome great struggles to achieve their goals, the more hopeful they will be [48]. Therefore, it could be that, for students who identify negative life experiences as giving rise to a sense of hope, they have developed a solidified sense of the strength of their self-agency and their ability to cope with adversity. It is likely, however, that this ability to be hopeful in challenging circumstances is predicated on these individuals' sense of potential mastery over their environment [48]. This is linked to the transcendent quality that was apparent in the conceptualisation of hope as underlying students' construal of their self over time in this present study. The ability to visualise a hoped-for future self allowed students to make sense of their past experiences and to cope with adversity in the present moment, engaging in goal-directed action to work towards the desired future. This perspective of hope aligns with Herth's description of hope as facilitating transcendence of the present situation and movement toward a new awareness and enrichment of being [50]. Herth's mode here is slightly more expansive than Snyder's, in which the transcendent quality of hope is not clearly articulated [51]. Moreover, this conceptualisation of hope additionally clearly links to Markus and Nurius' work on self-construal. The present study supports key conclusions from this body of work in that future self-conceptualisations, including desired future selves, influence people's present-moment construals, their motivation, and their behaviours [52].

For students who provided data on this topic, hope and depression were construed as inversely associated. Students described depression as this antithesis of hope and something that arose in the context of failed hope. This aligns with Snyder's description of people with low hope as having more negative thoughts about themselves and the future, and lacking goals and the drive to pursue them [48]. It aligns too with the positioning of hope as a natural balancing force against depression [53]. In turn, this makes sense in the context of Beck's model of depression; comprising negative thoughts and the self and the future as central components [54]. Nonetheless, data were limited and lacked a breadth of representation across different student groups and specific elicitations of the directions and processes of influence between hope and depression.

The current study additionally supports the importance of social connectedness in students' level of hope. Students draw, in many ways, a sense of hope from positive relationships with friends and family members. This aligns with prior research that perceived social support is positively associated with higher levels of self-esteem and optimism, which in turn leads to higher levels of hope [55]. Moreover, research has found that the association between positive social support and hope can have an impact on mental health [56, 57]. In contrast, a lack of perceived social support and low levels of hope can result in poor mental health, including anxiety and depression [58]. Therefore, the facilitation of student belonging and social connectedness is highly important for their sense of hope and broader wellbeing. In addition, students identified that (even brief) interactions with individuals with whom they do not have a personal relationship can impact hope. This appears to be especially the case when these individuals are powerful, respected, or expert in relation to the student's desired future, with

specific examples of academic faculty provided. This aligns with evidence in schools that student-teacher interactions and relationships can affect student hope in a myriad of ways, with implications for students' academic performance and socio-emotional development [59]. Together, the articulations of hope as fundamental, interpersonal, and related to the construal of self align with the additional conceptualised facets of hope (respectively "whypower" and "wepower") described in a complex systems perspective analysis of hope [15]. This alignment supports the need, as outlined by Colla and collagues, to view hope through a lens of complexity. It perhaps follows to move to a consideration of hope as an emergent property or system that reflects a dynamic interplay between parts, as opposed to something that can be understood by breaking it down into constituent components [15].

Finally, in previous research, hope and optimism were commonly regarded as positively related and were sometimes used interchangeably by lay people [60]. In this review, there was evidence that some students construed of hope and optimism as very similar. Others emphasised the broader and more passive conceptualisation of optimism *versus* the more agentic and specific nature of hope. This aligns with Snyder's perspective that optimism and hope are linked, but that optimism reflects mental energy for goals and belief they will happen yet does not necessarily mean that individuals can identify or effect the specific means to reach them [48]. This is supported by empirical research which has found that hope predicts unique variance in positive outcomes compared to optimism [61]. Other research too suggests an unconditional belief in a better future, akin to optimism, actually can predict lower levels of resilience and hope [60]. Thus, hope enables young people to take ownership of their desired future, construe themselves in having a role in enacting it, and engage in goal-directed behaviour. In comparison, the more vague and passive nature of optimism may offer students less opportunity to anticipate difficulties and create strategies to cope with failure [60].

## Strengths and limitations

There are several limitations that need to be addressed in future research on hope for higher education students. The findings of this review may not be generalisable to all student populations, as some studies included in this review predominantly consisted of White or female students. It is also important to note that six studies did not report the ethnicity of their participants. This review used only English language research which itself may have limited our ability to include reports involving more diverse samples. Students from different cultural backgrounds and health needs may conceptualise hope differently [1, 62]. Therefore, future research should focus on diverse student populations from various cultural and social backgrounds, including those with chronic mental or physical health conditions. Moreover, it is unclear if (and how many) students were postgraduate, but the majority appear to be undergraduate students. Postgraduate students, especially doctoral students, have more autonomy and less support. Doctoral students occupy high stress and low status positions [63, 64], with extremely high levels of mental health problems [65]. Thus, understanding how hope is construed and enhanced in postgraduate student population reflects an important gap in knowledge.

In addition, there were some examples in the qualitative data collated of students struggling to articulate their conceptualisation of hope. It can be challenging for people to define complex constructs, whilst still being able to give examples of their manifestation in daily life [6]. As data collection tools and processes were variable, it is possible that they did not use helpful or adequate means, such as asking for examples, to elicit students' concepts of hopefulness. Another notable limitation is the relative lack of focus and relevant data pertaining to mental health. Available data pertained quite exclusively to experiences of depression. This is

important, for studies show that depression is a very prevalent problem in higher education student populations [14, 65]. Nonetheless, high rates of other mental health problems, such as anxiety, exist in higher education student populations. Mechanisms of association between hope and mental health problems, especially other than depression, represents an important remaining gap in knowledge.

The meta-synthesis method used in this study is a strength, for it has been recognised as a rigorous technique for synthesising qualitative research [66]. Meta-synthesis allows for a deeper understanding of a field of research by bringing together data from multiple studies to generate new insights [66]. The use of GRADE-CERQual is another strength [34], providing clear guidance on the level of confidence to place in findings from this review. This metric is being increasingly used as evidence in decision-making processes, such as healthcare policies, interventions, and guideline developments [34]. The GRADE-CERQual assessment in this review supports having moderate confidence in all findings. Within the assessments of methodological limitations, relevance, and coherence, there were one or two minor and one or two major concerns respectively. All other findings identified none or very little concerns.

## Implications and future directions

Starting university is a challenging time for students as they transition into a new environment; maintaining academic performance while building new relationships can be a struggle [67]. As this review suggests, students view hope as predominantly cognitive, involving self-agency and pathways thinking focused on personally meaningful goals. Students appeared to find value in identifying their goals and visualising the person they hope to be in the future. Strategies such as psychoeducation about hope, learning to set goals, and promoting a growth mindset have been found to increase resilience and hope [26, 68]. Such interventions can be offered in brief forms and facilitated by non-experts [26]. Thus, higher education institutions should offer hope-enhancing interventions as part of their package of student support initiatives. Peer support may be a useful vehicle by which to offer these interventions. Peer supporters already use different approaches to support students with their sense of identity, social support, and mental health [69, 70]. Higher education institutions should ensure that faculty training consists of a focus on how best to support students to develop and maintain a sense of self-agency [63,71]. Moreover, having a strong social support system is also crucial for students' hope and, more generally, their mental health and psychological well-being [55,72]. Higher education institutions should consider how best to enhance the sense of community and reduce the sense of loneliness. Interventions that enhance social connectedness, for example through increasing social interaction or establishing social support groups, appear to currently have the most favourable evidence in reducing loneliness [73].

New evidence is needed to fill the specific knowledge gaps around how marginalised groups of students, and those pursuing postgraduate study, construe hope. Creative means of data collection should be used to help students articulate what appears to be a complex and multidimensional construct. The elicitation of metaphorical representations of hope would likely be useful. Metaphor helps us to make concrete connections between abstract concepts and everyday experiences [74]. Moreover, metaphors actually structure our experiences and shape our behaviour [74]. There were some naturally occurring examples of metaphor use in the reviewed data. However, further research designed to specifically elicit figurative language would help increase understandings of the behavioural manifestations of hope and what influences goal-direct action. In addition, while there is research supporting the effectiveness of hope interventions in higher education, the body of work remains quite small. Large-scale robust evaluations of effectiveness are needed to ensure these interventions do work and can

be implemented, and in turn therefore to encourage widescale roll-out. Finally, students in the reviewed studies identified that their interactions with faculty can have influential impact on their hope. Therefore, in addition to a practice-based implication to train academics to help scaffold students' hopeful self-agency, research is needed to help unpick how academics' own sense of hope may influence the hope-enhancing (or otherwise) relationships they develop with students in higher education institutions.

## Conclusion

This systematic review explored how higher education students describe and experience hope. Students identified hope as largely within their control and shaped by their choices, but nonetheless influenced by life events, relationships, and interactions with others. Hope was described as a complex and multi-dimensional experience that requires conscious effort and practice to cultivate and sustain. It is goal-oriented and tied to envisioning and pursuing the ideal future self. Hope often is borne from negative experiences but is positive in nature and confers a sense of resilience. Higher education institutions should focus institutional practices on enhancing students' self-agency and sense of community and decreasing loneliness. Brief interventions could be effectively offered within this setting to increase student hopefulness. Future research is needed to support the implementation of such programmes and to explore the conceptualisation of hope in diverse and postgraduate student populations.

## Supporting information

**S1 Checklist.**
(DOCX)

**S1 Table. Mixed methods appraisal tool (MMAT) quality appraisal.**
(DOCX)

**S2 Table. GRADE-CERQual evidence profile and summary of qualitative findings.**
(DOCX)

## Author Contributions

**Conceptualization:** Clio Berry.

**Data curation:** Nishi Acharya.

**Formal analysis:** Clio Berry, Nishi Acharya, Lucie Crowter.

**Investigation:** Clio Berry, Nishi Acharya.

**Methodology:** Clio Berry, Nishi Acharya.

**Project administration:** Clio Berry, Nishi Acharya.

**Supervision:** Clio Berry.

**Validation:** Clio Berry, Nishi Acharya, Lucie Crowter.

**Writing – original draft:** Clio Berry, Nishi Acharya.

**Writing – review & editing:** Clio Berry, Nishi Acharya, Lucie Crowter.

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
