## [Decision Letter · Decision Letter 0]

4 Mar 2024

PONE-D-23-34501The light at the end of the tunnel? A systematic review of higher education student experiences of hopePLOS ONE

Dear Dr. Berry,

Thank you for submitting your manuscript to PLOS ONE. After careful consideration, we feel that it has merit but does not fully meet PLOS ONE’s publication criteria as it currently stands. Therefore, we invite you to submit a revised version of the manuscript that addresses the points raised during the review process. Please see the comments in the attached file. There are not many but please look at the grammar and try to make your sentences shorter.

Please include the following items when submitting your revised manuscript:PlA rebuttal letter that responds to each point raised by the academic editor and reviewer(s). You should upload this letter as a separate file labeled 'Response to Reviewers'.A marked-up copy of your manuscript that highlights changes made to the original version. You should upload this as a separate file labeled 'Revised Manuscript with Track Changes'.An unmarked version of your revised paper without tracked changes. You should upload this as a separate file labeled 'Manuscript'.If applicable, we recommend that you deposit your laboratory protocols in protocols.io to enhance the reproducibility of your results. Protocols.io assigns your protocol its own identifier (DOI) so that it can be cited independently in the future. For instructions see: https://journals.plos.org/plosone/s/submission-guidelines#loc-laboratory-protocols. Additionally, PLOS ONE offers an option for publishing peer-reviewed Lab Protocol articles, which describe protocols hosted on protocols.io. Read more information on sharing protocols at https://plos.org/protocols?utm_medium=editorial-email&utm_source=authorletters&utm_campaign=protocols.

We look forward to receiving your revised manuscript.

Kind regards,

Mary Diane Clark, PhD

Academic Editor

PLOS ONE

Journal Requirements:

3. We note that your Data Availability Statement is currently as follows: [The data were drawn from the referenced primary research reports.]

Authors do not need to submit their entire data set if only a portion of the data was used in the reported study

4. Please include a caption for figure 1.

Additional Editor Comments (if provided):

Interesting article, thank you for allowing us to review it. I struggled to find reviewers so and going to use my own with the one that I did get.

Overall the paper is fine. Please do check the length of your sentences as noted in the attached file. There are a few other issues in that attachment.

Reviewers' comments:

Reviewer's Responses to Questions

**Comments to the Author**

1. Is the manuscript technically sound, and do the data support the conclusions?

Reviewer #1: Yes

2. Has the statistical analysis been performed appropriately and rigorously? 

Reviewer #1: N/A

3. Have the authors made all data underlying the findings in their manuscript fully available?

Reviewer #1: Yes

4. Is the manuscript presented in an intelligible fashion and written in standard English?

Reviewer #1: Yes

5. Review Comments to the Author

Reviewer #1: As an assistant professor working with undergraduate and graduate students in the field of Education of the Deaf and Hard of Hearing at a small university, I found your manuscript particularly relevant and engaging. My role involves extensive interaction with college students, and understanding their experiences, particularly in terms of hope and well-being, has a crucial aspect of my work and even more so post-Covid.

Your systematic review and meta-synthesis are impressively thorough. The methodological approach, including the extensive literature search and careful selection of studies, is admirable. The use of quality appraisal tools like MMAT and GRADE-CERQual further strengthens the reliability of your findings.

Given the qualitative nature of your study, the absence of detailed statistical analysis seems appropriate. The thematic analysis approach you've used effectively synthesizes the data and supports your conclusions in a meaningful way.

The Data Availability Statement in your manuscript is clear and adheres to the principles of transparency and accessibility, essential in research. Additionally, I found no issues regarding dual publication, research ethics, or publication ethics in your paper.

Your research significantly enriches our understanding of student experiences in higher education and is particularly meaningful to those of us dedicated to student education and welfare. Given my background, it would be fascinating to see similar research conducted specifically with Deaf and hard of hearing college students. Such studies could provide deeper insights into the unique experiences and challenges faced by this student population.

The process has been highly informative and a valuable learning experience for me. I've gained new insights into the field of higher education student experiences, which I can apply in my own academic and professional endeavors. This review has not only broadened my understanding of the subject matter but also enhanced my skills in critical analysis and evaluation of research work.

Thank you for the opportunity to review this work.

6. PLOS authors have the option to publish the peer review history of their article (what does this mean?). If published, this will include your full peer review and any attached files.

Reviewer #1: **Yes: **Dr. Frances F. Courson

---

## [Editor Report · Decision Letter 1]

15 May 2024

The light at the end of the tunnel? A systematic review of higher education student experiences of hope

PONE-D-23-34501R1

Dear Dr. Berry,

We’re pleased to inform you that your manuscript has been judged scientifically suitable for publication and will be formally accepted for publication once it meets all outstanding technical requirements.

Kind regards,

Mary Diane Clark, PhD

Academic Editor

PLOS ONE

Additional Editor Comments (optional):

Thanks for taking care of those corrections. At this point I am recommending that you are given the green light to go ahead. Congraduations